# Examining therapeutic equivalence between branded and generic warfarin in Brazil: The WARFA crossover randomized controlled trial

Carolina Gomes Freitas[1]*, Michael Walsh[2,3], Enia Lucia Coutinho[4], Angelo Amato Vincenzo de Paola[4], Álvaro Nagib Atallah[5]

1 Department of Medicine, Escola Paulista de Medicina, Universidade Federal de São Paulo (UNIFESP), São Paulo, SP, Brazil, 2 Departments of Medicine (Nephrology) and Health Research Methods, Evidence and Impact, McMaster University, Hamilton, ON, Canada, 3 Population Health Research Institute, McMaster University/Hamilton Health Sciences, Hamilton, ON, Canada, 4 Clinical Cardiac Electrophysiology, Cardiology Division, Department of Medicine, Federal University of Sao Paulo, Escola Paulista de Medicina, Sao Paulo, SP, Brazil, 5 Discipline of Emergency Medicine and Evidence-Based Medicine, Escola Paulista de Medicina, Universidade Federal de Sao Paulo (UNIFESP), São Paulo, SP, Brazil

* carolinagfreitas@outlook.com

## Abstract

### Objectives

To determine whether the generic and branded warfarins used as anticoagulants in Brazil are therapeutic equivalents based on their international normalized ratio (INR) results.

### Methods

This crossover randomized controlled trial had four periods. We used the branded Marevan and two generic versions of warfarin sodium tablets, manufactured by União Química and Teuto laboratories, all purchased from retail drugstores. Eligible participants were outpatients from an anticoagulation clinic at a university hospital in São Paulo, Brazil. They had atrial fibrillation or flutter and had been using warfarin for at least 2 months with an INR therapeutic range of 2.0–3.0. Randomization was by numbered, opaque, sealed envelopes. Healthcare personnel and outcome assessors were blinded to treatments, but patients were not. The primary outcome was the variability in the INR (ΔINR) and secondary outcomes included mean INR. We accepted formulations as equivalent if the 95% confidence interval (CI) of the comparison of ΔINR between branded and generic formulations was within the limit of ±0.49.

### Results

One hundred patients were recruited and randomized to six sequences of treatment (four sequences with n = 17 and two sequences with n = 16). União Química generic warfarin had equivalent variability in the INR to Marevan (ΔINR +0.09 [95% CI -0.29 to +0.46], n = 84). Comparison between Teuto generic warfarin and Marevan was inconclusive (ΔINR +0.29 [95% CI -0.09 to +0.68], n = 84).

**Data Availability Statement:** All relevant data are within the paper and its Supporting Information files.

**Funding:** ANA has received the Regular Research Grant 13/22618-6 from The São Paulo Research Foundation (http://www.fapesp.br/en/) that supported this study. The São Paulo Research Foundation (FAPESP) approved the design of the study but has not influenced: data collection and analysis, decision to publish nor preparation of the manuscript.

**Competing interests:** The authors have declared that no competing interests exist.

## Conclusions

Marevan and União Química warfarin had equivalent therapeutic effectiveness and both could be confidently used for anticoagulation. The comparison between Marevan and TW was inconclusive and does not warrant a statement of equivalence. Our methods are especially important for comparing generic and branded drugs that raise concerns and may be subject of future investigations by regulatory agents.

## Trial registration

ClinicalTrials.gov NCT02017197.

## Introduction

Worldwide, there is controversy over whether the anticoagulant effects of generic warfarin are equivalent to those of their branded counterparts [1–7]. Generic drugs do not routinely have their clinical or pharmacodynamic effects evaluated by health surveillance agencies: only bioequivalence outcomes are assessed, i.e., pharmacokinetic endpoints [8]. In addition, warfarin is a narrow therapeutic index (NTI) drug [9, 10], i.e., its effects must be rigorously monitored in order to balance benefits and risks.

After several years of international experience with generic drugs, the United States Food and Drug Administration [11], Health Canada [12], the European Medicines Agency [13], and the World Health Organization [14] have issued stricter rules for the therapeutic equivalence of generic versions of NTI drugs. These additional requirements have not yet been adopted by Brazil.

The INR effect of generic warfarin has been previously addressed in clinical trials [3–7], but never with drugs used in Brazil. As the formulation and manufacturing processes of drugs are highly influential in their bioavailability and, by extension, in their effect, the published data are not necessarily applicable to our context. Thus, we conducted the WARFA study, a crossover randomized controlled trial (RCT) comparing the effect on INR of Brazilian-branded warfarin with its generic versions for therapeutic equivalence.

## Material and methods

WARFA is registered in ClinicalTrials.gov (NCT02017197, URL https://clinicaltrials.gov/show/NCT02017197). Further details on the protocol and the statistical analysis plan of the WARFA trial have been described elsewhere [8]. WARFA was a single-center randomized crossover trial. The trial featured three drugs (allocation ratio 1:1:1), four periods and six potential sequences of study drugs to which participants were randomly allocated (S1 Fig). Each warfarin was taken for a one-month period, except for the first warfarin in the sequence, which was also used in the second period of the study. There was no washout period (with no treatment) for safety reasons. However, the INR measures were taken at the end of each treatment period in order to allow for the systemic elimination of the previous warfarin, thus avoiding its effects (carryover effect) [8].

The crossover design was chosen for this trial due to the interventions, and thus their expected outcomes, being very similar. In these cases, a crossover trial is more efficient, requiring a smaller sample size than a parallel design [15, 16].

Participant recruitment took place between August 2014 and March 2016, and follow up continued up to August 2016 in an outpatient anticoagulation clinic at the university hospital of the Universidade Federal de São Paulo, Brazil. The study has conformed to the ethical guidelines of the Declaration of Helsinki, the Brazilian guidelines on research involving human subjects [17], and was approved by the research ethics committee of the Universidade Federal de São Paulo. Freely given and informed written consent was obtained from all patients prior to inclusion in the trial.

Eligible patients met all the following criteria: aged at least 18 years; atrial fibrillation (AF) and/or atrial flutter (AFL) in the absence of a mechanical heart valve; $CHA_2DS_2VASC$ score of 1 or more [18]; on warfarin for at least 2 months before randomization; and able to provide written informed consent. Patients were excluded if any of the following applied: contraindication for warfarin; pregnancy, breastfeeding or women of childbearing potential; severe thrombocytopenia; advanced hepatic or renal failure; previous major bleedings due to congenital deficiency of coagulation factors; participation in a concurrent clinical trial; or they started chronic treatment with drugs that may potentially interact with warfarin, either affecting the INR, and/or the warfarin dose, and/or the risk of bleeding.

Initially, our inclusion criteria were limited only to subjects with nonvalvular AF. However, due to the difficulty of recruiting participants, we broadened our inclusion criteria to patients with AF and/or AFL without mechanical prosthetic valves. All of these patients are anticoagulated with warfarin to the same extent, i.e., with the goal INR in the same therapeutic range, so we do not expect this will bias our estimates of the effect of the warfarins [8].

After randomization, participants continued in the study if they had: 1) at least one of the three INR results in the first period within the therapeutic range and 2) a difference no greater than 0.8 in the INR results at the 3rd and 4th weeks. This measure was taken to reduce the variability in the INR values created by non-drug sources (e.g., patients with variable vitamin K intake), thus increasing the likelihood of detecting a true difference between warfarin formulations.

Three formulations of 5 mg warfarin sodium tablets produced in Brazil were used in the study: the branded Marevan (Farmoquímica, Rio de Janeiro) and the generic warfarins manufactured by União Química Farmacêutica Nacional (UQW), in Brasília, and Laboratório Teuto Brasileiro (TW), in Anápolis. All drugs were purchased from retail drugstores. Patients were instructed to take the warfarin tablets at the same time every day. Study visits, to monitor the outcomes and, if needed, adjust the dose (S1 Appendix) to maintain INR results within the therapeutic range (between 2.0 and 3.0) [19], were planned in the 3rd and 4th weeks of each period. Participants also received information from nutritionists regarding their intake of vitamin K food sources. All prothrombin time tests (and respective INRs) were processed by the same laboratory using the Stago-manufactured STA-R Evolution equipment and STA-Néoplastine CI Plus reagent (international sensitivity index 1.27).

We applied blocked randomization with a fixed block size of six. CGF enrolled patients and randomly allocated them by numbered, opaque, and sealed envelopes. Study drugs were in opaque sealed mailing bags of identical appearance except for a blinded assignment code, thus blinding study personnel. INR tests were also performed by blinded personnel at the hospital laboratory. Participants however, were not blinded to their treatment [8].

Our primary outcome was INR variability (ΔINR), defined as the absolute difference in the INRs recorded at the 3rd and 4th weeks of each study period, i.e., with the same warfarin formulation. Larger ΔINR suggest a more variable effect of a warfarin formulation and, therefore, a greater risk of INRs that are outside of the therapeutic range. The mean INR was a secondary outcome intended to assess whether formulations had consistently lower (or higher) INR

results when compared to each other, which could also pose the risk of out-of-the-therapeutic-range INRs.

We had equivalence hypotheses for the INR variability and the mean INR. Equivalence was met if the 95% CI of the difference between warfarin formulations in the ΔINR did not exceed ±0.49; the same criteria and equivalence limits were applied to the mean INR outcome. Each generic was compared with Marevan independently and then also to one another.

We considered that, for a patient with an average INR of 2.5, a change of 0.5 in the mean INR would probably be the minimum amount that would support a dose change. Variations in the mean INR of up to 0.49 could be dismissed as not clinically important, establishing thus the limits of the equivalence margin. Though we determined -0.49 to +0.49 as the equivalence margin for the mean INR, we decided to apply it to the INR variability outcome as well.

We had inequality hypothesis for all other secondary outcomes. These included: the mean warfarin dose needed per week; the Δdose (the absolute difference between the two weekly warfarin doses registered in each period, i.e., with the same drug formulation); and the mean time in therapeutic range (TTR), calculated using the Rosendaal and colleagues method [20, 21]. Clinical events and adherence were also recorded (definitions provided on S3 Appendix). All outcomes were assessed at the 3rd and 4th weeks of each study period.

We decided to change some trial outcomes at the statistical analysis plan stage. At first, the mean INR was our primary outcome. However, we considered that being an average between only two values, the mean INR would not be sensitive enough to detect important variations in each warfarin effect. We then included the ΔINR to address this concern, and the mean INR was changed to a secondary outcome. We also added the Δdose as a secondary outcome, again due to a concern regarding sensitivity of the mean dose outcome. These and other changes to the study protocol are described and explained in the S3 Appendix.

We considered 2.5 mg and 11.2%, respectively, as the minimally important differences in the dose-related outcomes and in the TTR outcome. The former represents the addition or withdrawal of at least a half warfarin tablet in the participants' weekly dose; the latter, the amount of change in the TTR that has been shown to dramatically alter warfarin efficacy [22].

Sample size was calculated to detect a clinically significant difference of 0.49 in the mean INR, assuming a significance level of 5% (two-sided), 90% power, and a standard deviation of 0.34 for the INR. The result was 11 participants in each group, a total of 33 for 3 warfarin formulations. This estimate assumed independent observations. Because in our study the observations were dependent, we expected this calculation to overestimate the sample size needed. However, due to the high rate of subjects not meeting the criteria for INR stability in the first period of the trial, we aimed to recruit 100 participants [8].

We analyzed data with STATA/IC 14.0. Comparisons were made using multilevel mixed-effects linear regression models with individuals as random intercepts. As the study design was crossover, and thus subject to carryover and other effects, we used a two-staged procedure to decide whether to analyze all the data (within-subjects) or only data from the first period as a parallel group trial (between subjects) [23]. We first modeled data, including the variables for sequence, period, and carryover; if any of these were significant at the pre-specified 5% level, then we analyzed only data from the first period [8, 24]. When none of these terms were significant, they were dropped from the model and data from all periods were considered in the analysis.

Considering that in an attempt to reduce the amount of variability in the INR values created by non-drug sources, participants with large changes in the INR in the first period were excluded from further randomized treatments, we analyzed three populations: The Complete cases, the First treatment period group and the Modified intention-to-treat (S2 Fig).

The First treatment period group population included only data of the first period for participants with at least one valid INR or ΔINR in this same period (S3 Appendix); this can be interpreted as an effectiveness analysis. The Complete cases population comprised data from participants that had at least one valid INR or ΔINR in every treatment period; it can be understood as an efficacy analysis. The Modified intention-to-treat population was composed of participants with at least one valid INR or ΔINR in any time period; it has mixed efficacy and effectiveness features. We pre-specified that if any of these analyses demonstrated non-equivalence between warfarin formulations, then we would consider that specific formulation non-equivalent. We did not impute missing outcome data.

The two-staged procedure showed that not all outcomes from the Complete cases and the Modified intention-to-treat populations were affected by period or sequence effects. Thus, as an exploratory sensitivity analysis, we also estimated the outcomes for these populations excluding the affected data when necessary. For instance, period effects were detected in the mean dose outcome of the Complete cases population at the 4th period; therefore, we estimated results for this outcome with data from the three previous periods.

Outcomes from the First treatment period group population that did not have repeated measures were analyzed with multiple linear regression models. We performed these analyses with baseline values as covariates [25] and their results are presented when supporting conclusions that are different from the unadjusted analyses. Adjusting the outcomes of this population for baseline values was not originally planned in our published protocol [8]. However, *post hoc*, we considered that the outcomes would depend on the participant's baseline values of INR and warfarin dose and, thus, decided for the adjustment.

We compared our results with those calculated by the bootstrap method (with 100 replicates) as an exploratory sensitivity analysis. Bootstrapping is a simulation procedure that may be used to assess the uncertainty of sample estimates [26]. We applied it due to concerns regarding the validity of the linear regression models used (the residuals did not fit into a normal distribution).

When comparing generic formulations, we report results of TW relative to UQW. We considered 95% confidence intervals significant for all comparisons and did not adjust for multiple comparisons.

## Results

The study population comprised 100 randomized participants (Table 1). The majority was male, with AF, high risk of stroke [18], and moderate risk of bleeding [27, 28]. Fig 1 describes the participant flow in the study.

Table 2 contains the results for the First treatment period group population. The analyses with baseline values as covariates only meaningfully changed the results of the mean dose outcome. The adjusted results for this outcome, which are expected to be more accurate, are presented as the primary results.

In the First treatment period group population, the CIs did not exceed -0.49 or + 0.49 and thus UQW showed equivalence to Marevan in the outcomes of INR variability and mean INR. The comparison between TW and the other formulations, both Marevan and UQW, was inconclusive for both INR-related outcomes: differences greater than +0.49 could not be excluded by the CI.

We have not found differences between the warfarins in the dose-related outcomes. The result of the Δdose outcome suggests that TW may have needed a significantly greater dose adjustment between the 3rd and 4th weeks than UQW. However, the Δdose of TW is very

**Table 1. Baseline characteristics of the patients included in the WARFA trial by treatment allocation in the first period of the study.**

| | Marevan (n = 33) | UQW (n = 34) | TW (n = 33) |
|---|---|---|---|
| **Age (years)**, mean (SD) | 68.1 (10.3) | 64.1 (10.1) | 66.3 (8.8) |
| **Female**, n (%) | 16 (48.5) | 13 (38.2) | 9 (27.3) |
| **Atrial fibrillation**, n (%) | 31 (93.9)[a] | 30 (88.2)[b] | 31 (93.9)[a] |
| Valvular AF, n (%) | 0 (0.0) | 2 (2.0) | 0 (0.0) |
| **Atrial Flutter**, n (%) | 4 (12.1)[a] | 5 (14.7)[b] | 4 (12.1)[a] |
| Valvular AFL, n (%) | 0 (0.0) | 1 (1.0) | 1 (1.0) |
| **CHA$_2$DS$_2$VASc**, mean (SD) | 3.4 (1.6) | 3.3 (1.4) | 3.1 (1.5) |
| **CHA$_2$DS$_2$VASc**, n (%) | | | |
| 0 | 1 (1.0) | 0 (0.0) | 1 (1.0) |
| 1 | 1 (3.0) | 2 (5.9) | 4 (12.1) |
| $\geq$ 2 | 31 (93.9) | 32 (94.1) | 28 (84.8) |
| **HAS-BLED**, mean (SD) | 1.5 (1.1) | 1.3 (1.1) | 1.5 (1.0) |
| **HAS-BLED**, n (%) | | | |
| 0 | 6 (18.2) | 9 (26.5) | 5 (15.1) |
| 1–2 | 22 (66.7) | 20 (58.8) | 23 (69.7) |
| $\geq$ 3 | 5 (15.1) | 5 (14.7) | 5 (15.1) |
| **CHF or LV dysfunction**, n (%) | 9 (27.3) | 13 (38.2) | 11 (33.3) |
| **Hypertension**, n (%) | 32 (97.0) | 32 (94.1) | 32 (97.0) |
| **Diabetes mellitus**, n (%) | 6 (18.2) | 7 (20.6) | 11 (33.3) |
| **Stroke**, n (%) | 5 (15.1) | 4 (11.8) | 3 (9.1) |
| **TIA**, n (%) | 0 (0.0) | 1 (2.9) | 1 (3.0) |
| **TE**, n (%) | 2 (6.1) | 2 (5.9) | 0 (0.0) |
| **MI**, n (%) | 6 (18.2) | 7 (20.6) | 4 (12.1) |
| **PAD**, n (%) | 2 (6.1) | 5 (14.7) | 1 (3.1) |
| **INR**, mean (SD) | 2.47 (0.67) | 2.48 (0.68) | 2.45 (0.65) |
| **Warfarin dose (mg) per week**, mean (SD) | 30.3 (10.1)[c] | 29.2 (12.7) | 33.1 (14.4) |

AF: atrial fibrillation; AFL: atrial flutter; CHF: congestive heart failure; LV: left ventricular; MI: myocardial infarction; PAD: peripheral artery disease; SD: standard deviation; TW: Teuto warfarin; TE: Thromboembolism; TIA: transient ischemic attack; UQW: União Química warfarin.

[a] 2 patients with both AF and AFL.

[b] 1 patient with both AF and AFL.

[c] n = 31 for this variable.

similar to that of Marevan, suggesting a slightly superior performance of UQW when compared with the other warfarins instead of any disadvantage for TW.

Regarding the mean time in therapeutic range, no difference was found between UQW and Marevan, whereas TW compared with the other warfarins resulted in lower time in the therapeutic range. However, we lack sufficient precision to determine the clinical interpretation of these results.

We have detected period and sequence effects for some outcomes in both the Complete cases and the Modified intention-to-treat populations (S1 Table). Generally, analyses with the Complete cases and the Modified intention-to-treat populations have shown equivalence between all three warfarins in the outcome of mean INR. We have not detected clinically meaningful differences between the formulations in the dose-related outcomes and, based on the minimally important difference considered, the outcome results of mean time in therapeutic range were inconclusive.

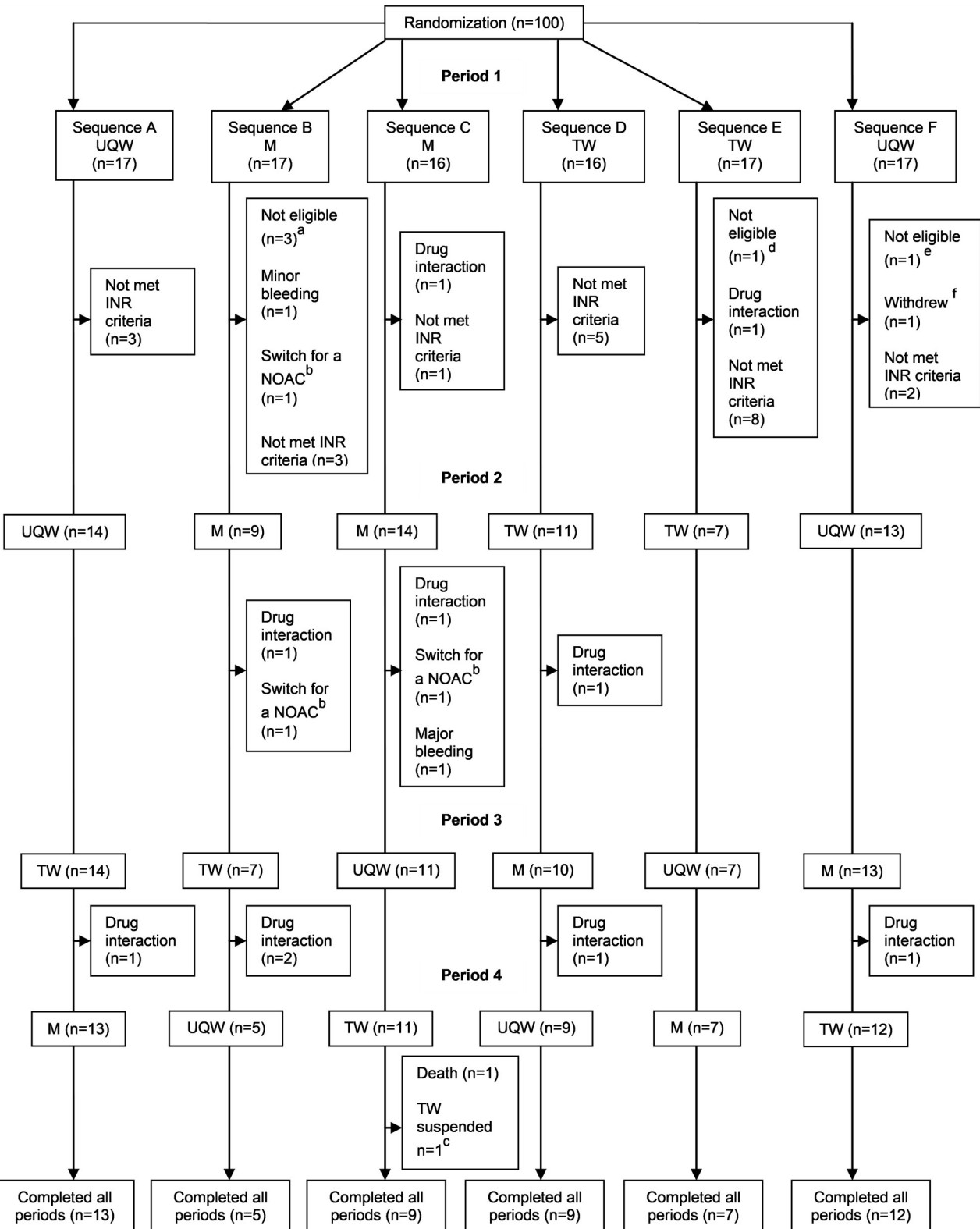

**Fig 1. Participant flow in the WARFA trial by allocation sequence and study period.** INR: international normalized ratio; M: Marevan; NOAC: novel anticoagulant; TW: Teuto warfarin; UQW: União Química warfarin. [a] One patient with $CHA_2DS_2VASc = 0$, one participant using warfarin along with aspirin and clopidogrel and one patient already enrolled in another clinical trial. [b] Warfarin switched for a NOAC due to arrhythmia

ablation procedures and not because of adverse events. [c] Patient developed hypersensitivity type I reaction to TW and decided to switch back to UQW. [d] One patient with CHA2DS2VASc = 0. [e] One patient of childbearing potential. [f] Patient withdrew due to study visits not fitting into his personal schedule.

The ΔINR was not estimated for these populations. This decision was due to the identification of period effects in the 2nd, 3rd and 4th periods of the Complete cases population. And in the Modified intention-to-treat population, because the interference of period effects in the 3rd and 4th periods of the trial led us to re-run the model without data from these periods, which resulted in the automatic omission of the treatment effects from the model due to collinearity.

The bootstrap simulation (S2 Table) generated results for the First treatment period group that were very similar to that obtained by analysis of the actual trial data, generally supporting the same conclusions. For the Complete cases and the Modified intention-to-treat populations however, the simulations encountered even more period and/or sequence effects than the original models and thus, not all outcomes were estimated. When the estimation of the outcome was possible after exclusion of the affected data, the results of the bootstrap simulations led to the same conclusions as the results from the actual trial data. Further results, including clinical events are available in the S4 Table.

## Discussion

Our main results, from the First treatment period group population, suggest that TW may result in clinically significant higher mean INR and more variability in the INR when compared with Marevan or UQW. On the contrary, UQW showed equivalence to Marevan in these same outcomes. Although the INR results could have been influenced by the adjustments in the warfarin dose that participants needed, the outcomes of Δdose and mean dose do not suggest differences that could leverage any of the warfarins. The time in therapeutic range results showed statistically lower performance for TW when compared with Marevan and UQW. However, the clinical meaning of all results for the TTR outcome is indeterminate because we were underpowered for this highly variable measure.

Carryover effects were not identified in our trial. Period and sequence effects were found in the analyses of some outcomes and populations, but their origins remain unclear. As previously stated in our published protocol [8], in the case that the results of the Complete cases and the Modified intention-to-treat populations were subject to sequence, period, or carryover effects, we would rely on the results of the First treatment period group population.

**Table 2. Outcomes of the WARFA trial for the First treatment period group population.**

| Outcome | Marevan Mean (SD) | UQW Mean (SD) | TW Mean (SD) | Differences between UQW and Marevan Mean (95%CI) | Differences between TW and Marevan Mean (95%CI) | Differences between TW and UQW Mean (95%CI) |
|---|---|---|---|---|---|---|
| ΔINR | 0.40 (0.30) | 0.49 (0.68) | 0.70 (0.93) | +0.09 (-0.29 to +0.46) | +0.29 (-0.09 to +0.68) | +0.20 (-0.16 to +0.57) |
| INR | 2.54 (0.65) | 2.49 (0.66) | 2.76 (1.11) | -0.05 (-0.41 to +0.30) | +0.23 (-0.12 to +0.59) | +0.29 (-0.06 to +0.64) |
| Dose (mg) per week | 28.1 (10.2) | 30.0 (13.0) | 33.5 (15.4) | +0.6 (-0.5 to +1.7)[a] | +0.9 (-0.2 to +2.0)[a] | +0.2 (-0.8 to +1.3)[a] |
| Δdose (mg) | 1.5 (3.3) | 0.3 (1.1) | 1.6 (3.3) | -1.2 (-2.6 to +0.3) | +0.1 (-1.4 to +1.6) | +1.3 (-0.1 to +2.7) |
| TTR (%) | 67.4 (40.5) | 71.2 (39.2) | 42.9 (40.1) | +3.7 (-17.6 to +25.1) | -24.5 (-46.3 to -2.6) | -28.2 (-48.9 to -7.5) |

ΔINR: INR variability; INR: international normalized ratio; SD: standard deviation; TW: Teuto warfarin; TTR: time in therapeutic range; UQW: União Química warfarin.

[a] Regression analyses of the mean dose per week outcome adjusted by the baseline dose. Unadjusted analyses resulted in a difference of +1.0 mg (95%CI -5.3 mg to +7.3 mg) between UQW and Marevan; +5.2 mg (95%CI -1.2mg to +11.6 mg) between TW and Marevan; and +4.1 mg (95%CI -2.1 mg to 10.4 mg) between TW and UQW.

Not all outcomes of the Complete cases and the Modified intention-to-treat populations were affected by these effects and thus, we also decided to estimate their results based on the data which were not affected. Conclusions for these populations were mostly the same as those obtained with the First treatment period group, with the exception of the outcome of mean INR. The Complete cases and the Modified intention-to-treat populations resulted in all warfarins showing equivalence on this outcome. Nevertheless, the bootstrap simulations for these populations, used to verify if results were sound, showed even more period and/or sequence effects than the original analyses, suggesting instability in the models used to calculate these results.

The condition of disagreement within the results of the different analytic populations was also anticipated in our protocol and, in that case, the most clinically important findings, i.e., non-equivalence, would be reported as the main results. Both conditions converge to the results of the First treatment period group population as being the primary in this trial.

One possible explanation for the divergence on the results of the First treatment period group and the other two populations, the Complete cases and the Modified intention-to-treat, might be the exclusion of patients with more variable INR results in the first period of the study. This decision may have inadvertently caused the INR results of the Complete cases and the Modified intention-to-treat populations to be more homogeneous than in the First treatment period group, thus resulting in no evidence of differences between the warfarin products.

As explained previously, we have selected patients with less variable INR in the first period of the trial in an attempt to 1) decrease INR variability caused by unstable vitamin K intake and 2) increase the chance of detecting true differences in the effects of the warfarins tested. A better approach, however, might be not trying to select patients in terms of vitamin K intake or other proxy, i.e., INR, especially considering that the crossover design, in which the patient is its own control, allows the results to account for patient's individual characteristics.

Our study has several strengths. Our main conclusions are protected from bias caused by post-randomization exclusions because they are based on the First treatment period group population, an effectiveness analysis that included data from all randomized participants. The results of the bootstrap simulation for the First treatment period group showed that our conclusions are robust despite initial concerns about the validity of the linear regression models applied. In addition, the adherence results suggest that the non-blinding of participants has not biased the outcomes by performance bias [29] because each patient adhered to the different warfarins to the same extent (S3 Table).

According to our literature review, an RCT comparing the INR effects of TW, UQW, and/ or Marevan has never before been conducted. In addition, our results are readily applicable in the anticoagulant treatments of nonvalvular AF/AFL patients. Aiming increased applicability of the results and prevention of conflict of interests, all drugs used in the study were purchased from retail pharmacies. Besides comparing the generic warfarins to Marevan, we also compared within generic formulations; even though they are not devised to be therapeutically equivalent, they are commonly switched in practice.

The drugs used in this RCT were the reference brand of warfarin sodium and the only two interchangeable generic versions approved by the Brazilian Health Regulatory Agency [30] at the time of study start. In the meantime, another brand of warfarin sodium, Marfarin, became interchangeable to Marevan [31] and the other warfarin brand, Coumadin, was discontinued [32]. Marfarin is also produced by União Química Farmacêutica Nacional in the same facility used to produce UQW [33], so it would be expected that the INR results of Marfarin may be similar to the ones obtained with UQW. Given that these are the only warfarins currently

available in Brazil, our study brings valuable information potentially applicable for all patients on anticoagulation with warfarin in the country.

We have not measured the drugs' concentration on patients' bloods, nor conducted bio-equivalence studies coupled with the WARFA trial, what could possibly have broadened the applicability of data to warfarin products with similar bioequivalence. In addition, in our study, clinical events (stroke, major bleedings, etc.) were secondary outcomes. Although they would be desirable primary outcomes, the necessary sample size would be much greater, threatening the feasibility of the study.

It also must be noted that our conclusions heavily rely on the equivalence margin selected for the INR-related outcomes. When the selected margin is inappropriately wide, we are at risk of stating equivalence between treatments that, in fact, are not equivalent. On the other hand, if the equivalence margin is too small, then a much larger sample size would be needed for the study to have adequate statistical power [34]. Our margin of equivalence was set based on the clinical criteria that we judged suitable.

The WARFA trial, along with previous crossover trials comparing generic and branded warfarins, has shown that applying an RCT for assessing the therapeutic equivalence of generic drugs using clinically meaningful endpoints is feasible. Indeed, due to the small differences expected between generic and branded drugs, the use of hard clinical endpoints (e.g., death, stroke, etc.) is unrealistic. Nevertheless, pharmacodynamic surrogate numeric endpoints (INR, blood pressure, etc.) coupled with suitable study designs (i.e., crossover) may increase the feasibility of using RCTs for these purposes and achieve adequate sample sizes in a reasonable amount of time.

Our methods are especially important for comparing generic and branded versions of narrow therapeutic index drugs, which were recently the subject of regulatory changes in many countries [11–13]. They are also useful for other drugs that raise concerns and may be subject of future investigations by regulatory agents. Pharmacodynamic data from RCTs would provide stronger evidence to back up theoretically and statistically based equivalence limits for generic drugs.

In conclusion, based on the outcomes of variability in the INR and mean INR, Marevan and UQW warfarins were therapeutically equivalent. Comparison between Marevan and TW was inconclusive, thus not warrant a statement of equivalence. According to our results, Marevan and UQW have equivalent therapeutic effectiveness and both can be confidently used for anticoagulation.

## Supporting information

**S1 Checklist. CONSORT 2010 checklist of information to include when reporting a randomised trial**∗.
(PDF)

**S1 Fig. Schema for the crossover design of the WARFA trial.** B, branded warfarin; A and C, generic formulations of warfarin, each from a different manufacturer. ∗Each period duration is one month. Republished with minor alterations from [8] under a CC BY license (http://creativecommons.org/licenses/by/4.0/), with permission from BioMed Central, original copyright 2017.
(TIF)

**S2 Fig. Populations for analysis in the WARFA trial.** A) First treatment period group population, which included only data of the first period for participants with at least one valid INR or ΔINR in this same period. B) Complete cases population, which comprised data from

patients that had at least one valid INR or ΔINR in every treatment period. C) Modified intention-to-treat population, which comprised participants with at least one valid INR or ΔINR in any time period. The arrows represent data collected from study subjects throughout the trial; each arrow represents one participant. The orange arrows represent the data from each patient that was included in that population for analysis.
(TIFF)

**S3 Fig. Flow diagram of the participants of the WARFA trial, by sequence and period, for the subpopulation First treatment period group and the outcomes of mean INR and mean warfarin dose per week.**
(PDF)

**S4 Fig. Flow diagram of the participants of the WARFA trial, by sequence and period, for the subpopulation First treatment period group and the outcomes of ΔINR, Δ dose, and mean TTR.**
(PDF)

**S5 Fig. Flow diagram of the participants of the WARFA trial, by sequence and period, for the subpopulation Modified intention-to-treat and the outcomes of mean INR and mean warfarin dose per week.**
(PDF)

**S6 Fig. Flow diagram of the participants of the WARFA trial, by sequence and period, for the subpopulation Modified intention-to-treat and the outcomes of ΔINR, Δ dose and mean TTR.**
(PDF)

**S7 Fig. Flow diagram of the participants of the WARFA trial, by sequence and period, for the subpopulation Complete cases and the outcomes of mean INR and mean warfarin dose per week.**
(PDF)

**S8 Fig. Flow diagram of the participants of the WARFA trial, by sequence and period, for the subpopulation Complete cases and the outcomes of ΔINR, Δ dose, and mean TTR.**
(PDF)

**S1 Table. Outcomes of the WARFA trial for the Complete cases and the Modified intention-to-treat populations.**
(PDF)

**S2 Table. Results for analyses of continuous outcomes of the WARFA trial by the bootstrap method (100 replicates).**
(PDF)

**S3 Table. Tenth percentile of the average adherence of participants to treatments by subpopulation of the WARFA trial (adherence in percentage).**
(PDF)

**S4 Table. Frequency of clinical events in the WARFA trial by warfarin product.**
(PDF)

**S5 Table. Baseline characteristics of the subpopulation First treatment period group, by sequence, for the outcomes of mean INR and mean warfarin dose per week.**
(PDF)

**S6 Table. Baseline characteristics of the subpopulation First treatment period group, by sequence, for the outcomes of ∆INR, ∆ dose, and mean TTR.**
(PDF)

**S7 Table. Baseline characteristics, by sequence and period, of the subpopulation Modified intention-to-treat for the outcomes of mean INR and mean warfarin dose per week.**
(PDF)

**S8 Table. Baseline characteristics, by sequence and period, of the subpopulation Modified intention-to-treat for the outcomes of ∆INR, ∆ dose, and mean TTR.**
(PDF)

**S9 Table. Baseline characteristics, by sequence and period, of the subpopulation Complete cases for the outcomes of mean INR and mean warfarin dose per week.**
(PDF)

**S10 Table. Baseline characteristics, by sequence and period, of the subpopulation Complete cases for the outcomes of ∆INR, ∆ dose, and mean TTR.**
(PDF)

**S1 Appendix. Protocol for adjustment of the warfarin dose applied in the WARFA trial.**
(PDF)

**S2 Appendix. Changes to the study protocol.**
(PDF)

**S3 Appendix. Definitions used in the WARFA trial.**
(PDF)

**S1 Protocol.**
(PDF)

**S2 Protocol.**
(PDF)

## Acknowledgments

CGF acknowledges the support of José Nunes de Alencar and Nelson Felipe Andrade, cardiologists at the Cardiac Electrophysiology Department at the Federal University Hospital of Sao Paulo, and Liz Andrea Kawabata Yoshihara, cardiologist at the University Hospital of the Sao Paulo University. We are also grateful to Beatrice Razzo and Maria Fernanda Santos Torres for their help in the randomization stage of the trial. We would also like to thank all the patients that participated in this trial.

## Author Contributions

**Conceptualization:** Carolina Gomes Freitas, Álvaro Nagib Atallah.

**Data curation:** Carolina Gomes Freitas.

**Formal analysis:** Carolina Gomes Freitas, Michael Walsh.

**Funding acquisition:** Carolina Gomes Freitas, Álvaro Nagib Atallah.

**Investigation:** Carolina Gomes Freitas.

**Methodology:** Carolina Gomes Freitas, Álvaro Nagib Atallah.

**Project administration:** Carolina Gomes Freitas.

**Resources:** Carolina Gomes Freitas, Enia Lucia Coutinho, Angelo Amato Vincenzo de Paola, Álvaro Nagib Atallah.

**Supervision:** Carolina Gomes Freitas, Álvaro Nagib Atallah.

**Visualization:** Carolina Gomes Freitas.

**Writing – original draft:** Carolina Gomes Freitas.

**Writing – review & editing:** Carolina Gomes Freitas, Michael Walsh, Enia Lucia Coutinho, Angelo Amato Vincenzo de Paola, Álvaro Nagib Atallah.

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
