## [Decision Letter · Decision Letter 0]

22 Dec 2020

PONE-D-20-22114

Examining therapeutic equivalence between branded and generic warfarin in Brazil: the WARFA crossover randomized controlled trial

PLOS ONE

Dear Dr. Gomes Freitas,

Thank you for submitting your manuscript to PLOS ONE. After careful consideration, we feel that it has merit but does not fully meet PLOS ONE’s publication criteria as it currently stands. Therefore, we invite you to submit a revised version of the manuscript that addresses the points raised during the review process.

We look forward to receiving your revised manuscript.

Kind regards,

Zhiming Zhu

Academic Editor

PLOS ONE

Journal Requirements:

2. We noted in your submission details that a portion of your manuscript may have been presented or published elsewhere:

'Fig 1 in this manuscript has been published along with the protocol of the study. Inclusion of the figure in this submission does not constitute dual publication because the figure only depicts the design of the trial, not disclosing any results. Inclusion of the figure in this manuscript is considered necessary to help readers understand the design of the trial.'

Please clarify whether this publication was peer-reviewed and formally published.

If this work was previously peer-reviewed and published, in the cover letter please provide the reason that this work does not constitute dual publication and should be included in the current manuscript.

Reviewers' comments:

Reviewer's Responses to Questions

**Comments to the Author**

1. Is the manuscript technically sound, and do the data support the conclusions?

Reviewer #1: Yes

Reviewer #2: Yes

2. Has the statistical analysis been performed appropriately and rigorously? 

Reviewer #1: Yes

Reviewer #2: Yes

3. Have the authors made all data underlying the findings in their manuscript fully available?

Reviewer #1: Yes

Reviewer #2: Yes

4. Is the manuscript presented in an intelligible fashion and written in standard English?

Reviewer #1: Yes

Reviewer #2: Yes

5. Review Comments to the Author

Reviewer #1: Comments to the Author 

Thank you for the opportunity to review your article. 

The authors report therapeutic equivalence between branded and generic warfarin in a therapeutic equivalence between branded and generic warfarin. Despite the work strengths, I recommend addressing the following comments: 

1. Did you monitor the concentration of branded and generic warfarin in every period?

2.S4 Table showed the clinical events in the WARFA trial, and could you record which period these clinical events first occur in? I think this table should be shown in the result section.

Reviewer #2: It is refreshing to see a conclusion on a non-significant result to be presented here in terms of the width of the confidence interval and not the lack of significance. The conclusions here are entirely sound given the data.

The point here is of course to rule out a difference and not detect a difference, as the positive result is a confidence interval that does not include the point of non-equivalence of 0.49. The sample size doesn't change because one assumes the drugs to be the same in efficacy. However this part of the design needs to be changed around somewhat as the trial is not a superiority trial. Given that the power calculation is for mean INR and not the new primary endpoint what were the assumptions for delta INR - and importantly what is the equivalence margin? Please explain why 0.49 is an appropriate equivalence margin here - why not 0.50?

It would be helpful to note earlier that the first two treatment periods are the same to motivate the fact there are only 6 permutations where every patient gets each drug.

The criteria for deciding equivalence or not could usefully be reiterated when the result is reported (i.e. CI does not include 0.49 ergo equivalent etc)

6. PLOS authors have the option to publish the peer review history of their article (what does this mean?). If published, this will include your full peer review and any attached files.

Reviewer #1: No

Reviewer #2: No

---

## [Author Response · Author response to Decision Letter 0]

11 Feb 2021

Response to reviewers

Journal Requirements

After checking my figures on PACE, I decided to improve figures 2 and 3. The new figures are uploaded. No other changes were made to the manuscript style because I believe that I have already followed the PLOS ONE's style requirements in my initial submission. In case I have missed any formatting requirement, please let me know so I can amend it accordingly.

2. We noted in your submission details that a portion of your manuscript may have been presented or published elsewhere:

'Fig 1 in this manuscript has been published along with the protocol of the study. Inclusion of the figure in this submission does not constitute dual publication because the figure only depicts the design of the trial, not disclosing any results. Inclusion of the figure in this manuscript is considered necessary to help readers understand the design of the trial.'

Please clarify whether this publication was peer-reviewed and formally published. If this work was previously peer-reviewed and published, in the cover letter please provide the reason that this work does not constitute dual publication and should be included in the current manuscript.

Figure 1 of this manuscript has already been formally published in a peer reviewed journal as part of the article entitled "Design and rationale for the WARFA trial: a randomized controlled cross-over trial testing the therapeutic equivalence of branded and generic warfarin in atrial fibrillation patients in Brazil". The previously published article has been provided in the submission process under the description of an "Other" file. As mentioned, the only item from this manuscript that has been published previously is the Figure 1, which was also included in this manuscript because it is deemed necessary for readers to clearly understand the design of the trial. This explanation was added to the cover letter and we added a note to the Figure 1 legend explaining that it is a reproduction from the previously published figure. Because this manuscript is not "a paper that overlaps substantially with one already published, without clear, visible reference to the previous publication" [1], it does not constitute dual publication.

Reviewer #1

1. Did you monitor the concentration of branded and generic warfarin in every period?

No, the concentration of warfarin in the participants' blood was not monitored. Unfortunately, for budget, time and complexity issues no pharmacokinetic parameter was measured, so, as mentioned in the manuscript, bioequivalence analyzes were not done. Also, for safety reasons, the warfarin dosage was adjusted during the trial as needed, according to the protocol in the S1 Appendix. However, knowing that the adjustment of dose could also interfere in the INR results, the primary outcomes in this trial, the doses were also recorded and analyzed in the form of two outcomes: mean dose and ∆dose. If the outcomes related to the doses suggested differences between the warfarin products, the INR results would have to be interpreted in light of this confounding factor.

2.S4 Table showed the clinical events in the WARFA trial, and could you record which period these clinical events first occur in? I think this table should be shown in the result section.

The information on when the first occurrence of the clinical events happened was added to S4 Table. Given that our primary outcomes are related to the INR and that we were probably underpowered to detect any difference on the occurrence of the clinical events between warfarin products, we decided to mention S4 Table on the results section of the manuscript, but leave it as supporting information.

Reviewer #2

The point here is of course to rule out a difference and not detect a difference, as the positive result is a confidence interval that does not include the point of non-equivalence of 0.49. The sample size doesn't change because one assumes the drugs to be the same in efficacy. However this part of the design needs to be changed around somewhat as the trial is not a superiority trial. Given that the power calculation is for mean INR and not the new primary endpoint what were the assumptions for delta INR - and importantly what is the equivalence margin? Please explain why 0.49 is an appropriate equivalence margin here - why not 0.50?

As mentioned on the manuscript (line 148), the equivalence margin adopted for the INR variability (∆INR) was also -0.49 to +0.49. The 95% confidence interval (CI) of the difference between warfarin formulations in their ∆INR could not exceed these values.

The rationale behind the equivalence margin is as follows: The therapeutic range for the INR for AF patients without a mechanical heart valve is from 2.0 to 3.0. Staying in the middle of the range, i.e., INR close to 2.5, is a goal that many healthcare providers would support, so as to avoid INRs outside of the therapeutic range. Considering a patient with an initial average INR of 2.5, a 0.5 change in their mean INR, either to an average INR of 2.0 or 3.0, would motivate a dose change. This was considered to be the minimum change in the mean INR of clinical significance, given it would motivate a dose change. Thus we considered that a reasonable equivalence margin for the mean INR would have to be immediately smaller, from -0.49 to +0.49.

This was the rationale behind the equivalence margin for the mean INR outcome; however, when we decided to add the outcome of INR variability (∆INR), we decided to apply the same equivalence margin.

Of note, even though we defined our equivalence margin for the INR outcomes from -0.49 to +0.49, our definition stated that for a conclusion of equivalence the 95% CI calculated for the outcomes could not exceed, i.e., could still include, these values. That is the same as an equivalence margin from -0.50 to +0.50 as defined by reviewer #2, given that according to the reviewer, these would be the points of non-equivalence and thus, 95%CI up to -0.49 or +0.49 would still be considered equivalent.

It would be helpful to note earlier that the first two treatment periods are the same to motivate the fact there are only 6 permutations where every patient gets each drug. The criteria for deciding equivalence or not could usefully be reiterated when the result is reported (i.e. CI does not include 0.49 ergo equivalent etc)

The interpretation of results based on the CI and its position in relation to the equivalence margin was added to the results section of the manuscript.

References

1. International Committee of Medical Journal Editors. Overlapping Publications [Internet]. [place unknown]: International Committee of Medical Journal Editors; c2021 [cited 2021 Feb 03]. Available from: http://www.icmje.org/recommendations/browse/publishing-and-editorial-issues/overlapping-publications.html

---

## [Decision Letter · Decision Letter 1]

2 Mar 2021

Examining therapeutic equivalence between branded and generic warfarin in Brazil: the WARFA crossover randomized controlled trial

PONE-D-20-22114R1

Dear Dr. Gomes Freitas,

We’re pleased to inform you that your manuscript has been judged scientifically suitable for publication and will be formally accepted for publication once it meets all outstanding technical requirements.

Kind regards,

Zhiming Zhu

Academic Editor

PLOS ONE

Additional Editor Comments (optional):

Reviewers' comments:

Reviewer's Responses to Questions

**Comments to the Author**

1. If the authors have adequately addressed your comments raised in a previous round of review and you feel that this manuscript is now acceptable for publication, you may indicate that here to bypass the “Comments to the Author” section, enter your conflict of interest statement in the “Confidential to Editor” section, and submit your "Accept" recommendation.

Reviewer #1: All comments have been addressed

Reviewer #2: All comments have been addressed

2. Is the manuscript technically sound, and do the data support the conclusions?

Reviewer #1: Yes

Reviewer #2: (No Response)

3. Has the statistical analysis been performed appropriately and rigorously? 

Reviewer #1: Yes

Reviewer #2: (No Response)

4. Have the authors made all data underlying the findings in their manuscript fully available?

Reviewer #1: (No Response)

Reviewer #2: (No Response)

5. Is the manuscript presented in an intelligible fashion and written in standard English?

Reviewer #1: (No Response)

Reviewer #2: (No Response)

6. Review Comments to the Author

Reviewer #1: (No Response)

Reviewer #2: (No Response)

7. PLOS authors have the option to publish the peer review history of their article (what does this mean?). If published, this will include your full peer review and any attached files.

Reviewer #1: No

Reviewer #2: No

---

## [Editor Report · Acceptance letter]

16 Mar 2021

PONE-D-20-22114R1 

Examining therapeutic equivalence between branded and generic warfarin in Brazil: the WARFA crossover randomized controlled trial 

Dear Dr. Gomes Freitas:

I'm pleased to inform you that your manuscript has been deemed suitable for publication in PLOS ONE. Congratulations! Your manuscript is now with our production department. 

Kind regards, 

on behalf of

Dr. Zhiming Zhu 

Academic Editor

PLOS ONE